# Antiparasitic Meroterpenoids Isolated from *Memnoniella dichroa* CF-080171

**DOI:** 10.3390/pharmaceutics15020492

**Published:** 2023-02-02

**Authors:** Frederick Boye Annang, Guiomar Pérez-Moreno, Cristina Bosch-Navarrete, Victor González-Menéndez, Jesús Martín, Thomas A. Mackenzie, Maria C. Ramos, Luis M. Ruiz-Pérez, Olga Genilloud, Dolores González-Pacanowska, Francisca Vicente, Fernando Reyes

**Affiliations:** 1Fundación MEDINA, Centro de Excelencia en Investigación de Medicamentos Innovadores de Andalucía, Parque Tecnológico de Ciencias de la Salud, Avda. del Conocimiento 34, 18016 Granada, Spain; 2Instituto de Parasitología y Biomedicina “López-Neyra”, Consejo Superior de Investigaciones Científicas (CSIC) Avda. del Conocimiento 17, Armilla, 18016 Granada, Spain

**Keywords:** natural products, meroterpenoids, *Memnoniella*, parasites, tropical diseases, bioassay-guided isolation, HRMS, NMR, malaria, trypanosoma, *Plasmodium falciparum*, Chagas disease

## Abstract

*Memnoniella* is a fungal genus from which a wide range of diverse biologically active compounds have been isolated. A *Memnoniella dichroa* CF-080171 extract was identified to exhibit potent activity against *Plasmodium falciparum* 3D7 and *Trypanosoma cruzi* Tulahuen whole parasites in a high-throughput screening (HTS) campaign of microbial extracts from the Fundación MEDINA’s collection. Bioassay-guided isolation of the active metabolites from this extract afforded eight new meroterpenoids of varying potencies, namely, memnobotrins C-E (**1**–**3**), a glycosylated isobenzofuranone (**4**), a tricyclic isobenzofuranone (**5**), a tetracyclic benzopyrane (**6**), a tetracyclic isobenzofuranone (**7**), and a pentacyclic isobenzofuranone (**8**). The structures of the isolated compounds were established by (+)-ESI-TOF high-resolution mass spectrometry and nuclear magnetic resonance spectroscopy. Compounds **1, 2**, and **4** exhibited potent antiparasitic activity against *P. falciparum* 3D7 (EC_50_ 0.04–0.243 μM) and *T. cruzi* Tulahuen (EC_50_ 0.266–1.37 μM) parasites, as well as cytotoxic activity against HepG2 tumoral liver cells (EC_50_ 1.20–4.84 μM). The remaining compounds (**3**, **5**–**8**) showed moderate or no activity against the above-mentioned parasites and cells.

## 1. Introduction

Malaria and Chagas are two parasitic diseases with a combined global risk population of over a billion people, infecting about 248 million people per year and killing about 637,000 of them [1,2]. Malaria is caused by protozoan parasites of the genus *Plasmodium*, and the five species which affect humans include *P. falciparum*, *P. vivax*, *P. ovale*, *P. malariae* and *P. knowlesi* [1]. The *P. falciparum* and *P. vivax* species pose the greatest human threat, with the former being the most prevalent and virulent, accounting for over 90% of all reported cases [3]. Human infection and transmission of the parasites are caused by the bite of infected female Anopheles mosquitoes. Although a recent significant increase in malaria research funding (an estimated USD 3 billion in 2019) has led to sustained progress in the fight against the disease (44% drop in mortality from 2010 to 2019), emerging multi-drug resistant parasites continue to be a real threat towards the elimination and eventual eradication of this debilitating disease [1,3,4]. Chagas is also a protozoan parasitic disease caused by *Trypanosoma cruzi* [2,5]. This disease is most prevalent in Latin America and mainly transmitted to humans when a triatomine bug bites and introduces parasite-infected faeces or urine at the site of the bite [2,5]. Chagas disease manifests in two forms: an asymptomatic, relatively short (about two months) acute phase with high numbers of blood circulating parasites and a chronic latent parasitaemia phase, which, when untreated, can lead to cardiac, digestive, and neurological disorders decades later [2,5,6]. The anti-Chagas drugs, benznidazole and nifurtimox, are both effective against the acute stage but require long hospital stays, are ineffective against the chronic stage, are counter-indicated in pregnancy or kidney/liver failure patients, and have resistance as well as adverse side effect problems [2,5,6]. Both malaria and Chagas require the discovery and development of new compounds for their therapeutic intervention [1,2,4,6].

Natural products are nature-inspired small molecular weight secondary metabolites which offer an underexplored chemical space with abundant bioactive molecules awaiting their discovery and application to current medical needs. *Memnoniella* and *Stachybotrys* are two closely related fungal genera from the microbial part of this chemical space that have been extensively studied over the years for their rich endowment with a wide diversity of biologically active natural products [7]. From these two fungal genera alone, over 200 secondary metabolites of wide chemical diversity have been isolated [7], including trichothecenes [8], triprenyl phenols [9], diterpenoids [7,10], isochromanes [11], polyketides [12], cochlioquinones [13], cyclic peptides [14], griseofulvins [15] and others [16,17].

Recent high-throughput screening (HTS) of a subset of microbial extracts from the Fundación MEDINA’s collection against *P. falciparum* 3D7 and *T. cruzi* Tulahuen whole parasites led to the identification of a *Memnoniella dichroa* CF-080171 extract that exhibited potent activity against both parasites. Bioassay-guided fractionation (reversed-phase C-18 semi-preparative HPLC) of a dichloromethane crude extract of the fungus cultivated in a rice-based solid BRFT medium afforded eight new meroterpenoids, namely, memnobotrins C-E (**1**–**3**), a glycosylated isobenzofuranone (**4**), a tricyclic isobenzofuranone (**5**), a tetracyclic benzopyrane (**6**), a tetracyclic isobenzofuranone (**7**) and a pentacyclic isobenzofuranone (**8**). The work presented here describes the isolation, structure elucidation and bioactivity characterization of compounds **1**–**8** (Figure 1).

## 2. Materials and Methods

### 2.1. General Experimental Procedures

The optical rotations of compounds **1**–**8** were measured with a Jasco P-2000 polarimeter (JASCO Corp., Tokyo, Japan). The IR data of all the compounds were obtained with a JASCO FT/IR-4100 spectrometer equipped with a PIKE MIRacle single reflection ATR accessory (JASCO Corp., Tokyo, Japan). Their UV spectra were obtained with an Agilent 1100 DAD (Agilent Technologies, Santa Clara, CA, USA). The NMR data acquisition was with a 1.7 mm TCI MicroCryoProbe-equipped Bruker Avance III spectrometer (500 and 125 MHz for ^1^H and ^13^C NMR, respectively) from Bruker Biospin, Fällanden, Switzerland. The compounds were previously dissolved in CD_3_OD, and their NMR chemical shifts are reported in ppm. The NMR signals of the residual solvent were used as internal reference (δ_H_ 3.31 and δ_C_ 49.15 for CD_3_OD). The (+)-ESI-TOF mass spectra of **1**–**8** were acquired with a Bruker maXis QTOF mass spectrometer from Bruker Daltonik GmbH, Bremen, Germany [18,19]. Semipreparative HPLC separations and subsequent compound purifications were performed on Gilson GX-281 322H2 HPLC with UV detection at 210 and 280 nm (Gilson Technologies, Middleton, WI, USA). 

### 2.2. Producer Fungal Strain Isolation and Characterization 

The extract-producing fungal strain, *M. dichroa* CF-080171, was obtained from decaying plant material collected in Chile. For storage and preservation of the axenic strain, 10% glycerol suspensions of septate mycelium and conidia were prepared and frozen at −80 °C. This strain is currently maintained in the Fungal Culture Collection of Fundación MEDINA (http://www.medinadiscovery.com accessed on 9 December 2022). For strain characterization purposes, DNA extraction, PCR amplification and DNA sequencing were performed as previously described by Gonzalez-Menendez et al., 2017 [20]. Sequences of the complete ITS1-5.8S-ITS2 and initial 28S region or independent ITS and partial 28S rDNA were deposited in GenBank, with the accession number OP554575. The BLAST application was used in comparing this new sequence to those already existing in the GenBank (https://www.ncbi.nlm.nih.gov accessed on 9 December 2022) and NITE Biological Resource Center (http://www.nbrc.nite.go.jp/ accessed on 9 December 2022) databases [21,22]. Database matching with the ITS rDNA sequence yielded a complete sequence similarity (100%) to the strain of *Memnoniella dichroa* CBS 526.50 GenBank Accession No. KU846140, indicating the genetic similarity of strain CF-080171 to *M. dichroa* (i.e., the two strains are conspecific). Similarly, high matching scores were obtained with other authentic fungal strains of this species, e.g., *M. dichroa* ATCC 18913 (GenBank Accession No. AF081472, 100% sequence similarity) and *M. dichroa* strain CBS 123800 (GenBank Accession No. KU846141, 100% sequence similarity), thus confirming that CF-080171 can be classified as *Memnoniella dichroa* (Grove) L. Lombard & Crous [23].

### 2.3. Culturing Conditions for M. dichroa CF-080171 

For the culturing of *M. dichroa* CF-080171, ten mycelial agar plugs of the strain were inoculated into flasks of SMYA medium and incubated on a rotary shaker at 22 °C with 220 rpm and 70% relative humidity. After 7 days of growth, 4 mL aliquots of this seed culture were used to inoculate 100 mL of the production rice-based solid BRFT medium in 10 × 500 mL Erlenmeyer flasks and incubated under static conditions at 22 °C and 70% relative humidity for 21 days [24].

### 2.4. Bioassay-Guided Isolation of Compounds **1**–**8**

After 21 days of incubation, the 1 L fermented solid whole broth of *M. dichroa* CF-080171 was harvested, and 1 L of milliQ water was added to it to create a homogeneous aqueous suspension. This suspension was then extracted with 1 L acetone by shaking in a Kuhner shaker at 200 rpm, 24 °C, for 2 h. The extract was filtered in a vacuum flask, and the acetone subsequently evaporated under a stream of nitrogen gas (N_2_) to obtain a concentrated aqueous crude. An aliquot of the aqueous crude extract was tested and found to be active against both *P. falciparum* 3D7 and *T. cruzi* Tulahuen C4 parasite strains. The 1 L crude was then extracted with an equal volume of dichloromethane (DCM). The extraction was performed 2 times, after which the organic phases were combined and dried to yield 892 mg of crude that tested positive against both *P. falciparum* and *T. cruzi* parasites. This crude was re-dissolved in DMSO, filtered through a 0.2 µM microfilter and subjected to semi-preparative reversed-phase HPLC (linear gradient 5–100% acetonitrile/H_2_O in 45 min) using an Agilent Zorbax SB-C18 column (9.4 × 250 mm, 5 µm) at 3.6 mL/min flow rate with UV detection at 210 and 280 nm. Repeated injections of aliquots of the extract under these HPLC conditions, in combination with antiparasitic testing and LC/MS analysis of the fractions, identified the bioactive fractions of interest. Repurification of these fractions of interest with a linear gradient of 20–100% acetonitrile/H_2_O in 45 min using the same HPLC column, flow and detection conditions indicated above (or 20–70% acetonitrile/H_2_O in 45 min specifically for compounds **2** and **5**) led to the isolation of 0.8 mg of compound **1** (rt 21.5 min), 0.9 mg of **2** (rt 22.00 min), 1.0 mg of **3** (rt 25.0 min), 0.2 mg of **4** (rt 21.25 min), 0.85 mg of **5** (rt 22.45), 0.45 mg of **6** (rt 22.55 min), 0.3 mg of **7** (rt 27.00 min) and 0.3 mg of **8** (rt 22.75 min). The physical appearance, specific optical rotation, UV/IR absorption and mass spectra characterization of each of the compounds are indicated below (see Appendix A for detailed spectroscopic data). 

Compound **1**: yellowish amorphous solid; [α]_D_^24^ +147 (c 0.15, MeOH); UV (DAD) 210, 280 nm; IR (ATR) ν_max_ 3343, 2943, 2830, 1448, 1114 and 1022 cm^−1^, ^1^H and ^13^C NMR data, see Table 1, (+)-ESI-TOF *m/z* 428.2440 [M+H]^+^ (calcd. for C_25_H_34_NO_5_, 428.2432), 855.4792 [2M+H]^+^ (calcd for C_50_H_67_N_2_O_10,_ 855.4791) and 1282.7153 [3M+H]^+^ (calcd for C_75_H_100_N_3_O_15,_ 1282.7149).

Compound **2**: yellowish amorphous solid; [α]_D_^24^ +34 (c 0.15, MeOH); UV (DAD) 210, 280 nm; IR (ATR) ν_max_ 3333, 2944, 2830, 1403, 1105 and 1023 cm^−1^, ^1^H and ^13^C NMR data, see Table 1, (+)-ESI-TOF *m/z* 430.2595 [M+H]^+^ (calcd. for C_25_H_36_NO_5_, 430.2588), 859.5102 [2M+H]^+^ (calcd for C_50_H_71_N_2_O_10,_ 859.5104) and 1288.7658 [3M+H]^+^ (calcd for C_75_H_106_N_3_O_15,_ 1288.7619).

Compound **3**: yellowish amorphous solid; [α]_D_^24^ +38 (c 0.1, MeOH); UV (DAD) 210, 280 nm; IR (ATR) ν_max_ 3306, 2949, 2831, 1671, 1614, 1248, and 1016 cm^−1^, ^1^H and ^13^C NMR data, see Table 2, (+)-ESI-TOF *m/z* 472.2701 [M+H]^+^ (calcd. for C_27_H_38_NO_6,_ 472.2694) and 943.5318 [2M+H]^+^ (calcd for C_54_H_75_N_2_O_12,_ 943.5315).

Compound **4**: yellowish amorphous solid; [α]_D_^24^ −24 (c 0.1, MeOH); UV (DAD) 210, 280 nm; IR (ATR) ν_max_ 3415, 2935, 1720, 1615, 1455, 1351, 1246, 1062 and 1019 cm^−1^, ^1^H and ^13^C NMR data, see Table 3, (+)-ESI-TOF *m/z* (+)-ESI-TOF *m/z* 568.3131 [M+NH_4_]^+^ (calcd. for C_29_H_46_NO_10,_ 568.3117), 1118.5912 [2M+NH_4_]^+^ (calcd for C_58_H_88_NO_20,_ 1118.5895) and 371.2226 [M-C_6_H_12_O_6_ +H]^+^ (calcd. for C_23_H_31_O_4_, 371.2217).

Compound **5**: yellowish amorphous solid; [α]_D_^24^ +35 (c 0.06, MeOH); UV (DAD) 210, 280 nm; IR (ATR) ν_max_ 3303, 2908, 2826, 1740, 1617, 1336, 1243, 1186, 1084 and 1013 cm^−1^, ^1^H and ^13^C NMR data, see Table 3, (+)-ESI-TOF *m/z* 403.2118 [M+H]^+^ (calcd. for C_23_H_31_O_6,_ 403.2116), 385.2014 [M-H_2_O+H]^+^ (calcd for C_23_H_28_O_5,_ 385.2010) and 805.4163 [2M+H]^+^ (calcd. for C_46_H_61_O_12_, 805.4158).

Compound **6**: white amorphous solid; [α]_D_^24^ −18 (c 0.05, MeOH); UV (DAD) 210, 280 nm; IR (ATR) ν_max_ 2930, 2855, 1746, 1689, 1587, 1455 and 1339 cm^−1^, ^1^H and ^13^C NMR data, see Table 4, (+)-ESI-TOF *m/z* 345.2426 [M+H]^+^ (calcd. For C_22_H_33_O_3,_ 345.2425).

Compound **7**: white amorphous solid; [α]_D_^24^ −27 (c 0.03, MeOH); UV (DAD) 210, 280 nm; IR (ATR) ν_max_ 3303, 2946, 2831, 2680, 1748, 1693, 1614, 1345, 1246, 1196, 1141, 1084 and 1016 cm^−1^, ^1^H and ^13^C NMR data, see Table 4, (+)-ESI-TOF *m/z* 385.2014 [M+H]^+^ (calcd. For C_23_H_29_O_5,_ 385.2010) and 786.421 [2M+NH_4_]^+^ (calcd. For C_46_H_60_NO_10_, 786.4212).

Compound **8**: white amorphous solid; [α]_D_^24^ +117 (c 0.03, MeOH); UV (DAD) 210, 280 nm; IR (ATR) ν_max_ 3302, 2938, 2829, 1747, 1618, 1420 and 1014 cm^−1^, ^1^H and ^13^C NMR data, see Table 5, (+)-ESI-TOF *m/z* 387.2172 [M+H]^+^ (calcd. for C_23_H_31_O_5,_ 387.2166) and 790.4526 [2M+NH_4_]^+^ (calcd. for C_46_H_64_NO_10_, 790.4525).

The isolated compounds were then subjected to the biological assays described below to determine their antiparasitic and cytotoxic effects.

### 2.5. Biological Assays

#### 2.5.1. *P. falciparum* 3D7 Lactase Dehydrogenase Assay

The EC_50_ determinations of the isolated compounds were performed with 16-point dose-response curves. Each concentration was evaluated in triplicate (concentration range of 100–0.00305 µM) using the *P. falciparum* 3D7 lactate dehydrogenase whole parasite assay as previously described by Pérez-Moreno et al., 2016 [25]. The *Plasmodium falciparum* 3D7 strain is a clone of the NF54 strain, which was previously isolated from a patient [26]. The parasites were maintained in O^+^ human erythrocytes at 5% haematocrit and 0.1–8% parasitaemia, with incubation at 37 °C, 1% O_2_, 5% CO_2_, and 94% N_2_ in complete medium (10.4 g/L RPMI 1640, 150 μM hypoxanthine, 12.5 μg/mL gentamicin, 0.2% NaHCO_3_, 0.5% albuMAX II, 2% human O^+^ serum). The lactase dehydrogenase (LDH) assay was performed in 384-well plates using late-ring/early trophozoite synchronized parasites at 2% hematocrit and 0.25% parasitaemia. Twenty-five microliters of (25 µL) parasite culture were incubated with 5 µL of the test compounds (parasite medium/100 nM chloroquine used as positive/negative growth controls) for 72 h, after which the plates were frozen for 4 h, thawed for 1 h at room temperature before adding 70 µL of freshly prepared LDH solution (1 U/mL diaphorase, 143 µM APAD, 143 mM sodium L-lactate, 178.75 µM NBT, 0.7% Tween 20, 100 mM Tris-HCL) at pH 8.0. Plates were briefly shaken and incubated for 10 min (in the dark), and absorbance was read at 650 nm in an Envision plate reader (Perkin Elmer, Waltham, MA, USA).

The EC_50_ of all the biological assays were computed with the Genedata Screener application (Genedata AG, Basel, Switzerland) using the equation below to calculate percentage inhibition:$$\mathrm{Percentage}\;\mathrm{inhibition}=\left[1-\left(\frac{Abs_{well}-Abs_{neg}}{Abs_{pos}-Abs_{neg}}\right)\right]\times 100$$

Here, *Abs_well_* = absorbance/well, *Abs_pos_* = average absorbance in positive control wells, and *Abs_neg_* = average absorbance in negative wells.

#### 2.5.2. Transgenic *T. cruzi* β-D-galactosidase Assay

The EC_50_ determinations of the isolated compounds were performed with 16-point dose-response curves. Each concentration was evaluated in triplicate (concentration range of 50–0.00152 µM) in the transgenic *T. cruzi* β-D-galactosidase assay as previously described by Annang et al., 2014 [27]. The *T. cruzi* Tulahuen strain used is a genetically modified strain that expresses the *Escherichia coli* β-galactosidase gene, lacZ [28]. It was kindly supplied by Marcel Kaiser (Swiss Tropical and Public Health Institute). The parasites were generally maintained in a growth medium (RPMI 1640 supplemented with 10% inactivated fetal bovine serum (iFBS), 100 µg/mL streptomycin, 100 U/mL penicillin, 2 mM L-glutamine) at 37 °C, 5% CO_2_. For the assay, transgenic trypomastigote stage parasites were used to infect L6 rat skeletal muscle cells (host). Fifty-five microliters (55 µL) of the infected L6 cell culture were dispensed into 384-well assay plates (2 × 10^3^ infected L6 cells/well) containing 5 µL of the test compounds (parasite medium/10 µgmL^−1^ benznidazole were used as positive/negative growth controls) and incubated for 96 h at 37 °C, after which 15 µL of assay solution (100 µM CRPG and 0.1% NP40 made up in PBS) were added to each well, and plates were further incubated in the dark for 4 h at 37 °C. Absorbance in the wells was read at 585 nm in an Envision plate reader (Perkin Elmer, Waltham, MA, USA). 

#### 2.5.3. MTT-Based Cytotoxicity Assay in HepG2 Cells

The EC_50_ determinations of the isolated compounds were performed with 12-point dose-response curves. Each concentration was evaluated in triplicate (concentration range of 50–0.0244 µM) in an MTT-based assay performed in liver carcinoma Hep G2 HB-8065 ATCC (American Type Culture Collection) cells as previously described by Annang et al., 2020 [29]. Cells (96-well plates were used with seeding at 1 × 10^4^ cells/well) were cultured in 200 µL MEM medium per well at 37 °C, 5% CO_2_ for 24 h, after which the spent media were replaced with 200 µL MEM medium and 1 μL of test compounds. DMSO (0.5%), 8 mM methyl methanesulfonate and doxorubicin were used as positive, negative, and standard drug controls, respectively. After 72 h incubation with the test compounds at 37 °C, 100 µL of 0.5 mg/mL MTT solution diluted in MEM without phenol red was added to each well, and plates were briefly shaken and further incubated for 3 h at 37 °C. The supernatant in each well was carefully discarded and replaced with 100 µL of DMSO, and plates were gently shaken (to solubilize the formazan formed) before reading the absorbance at 570 nm in a Victor2 plate reader (Perkin Elmer, USA).

#### 2.5.4. Cytotoxicity Assay in L6 Rat Skeletal Muscle Cells

L6 CRL-1458 ATCC rat skeletal muscle cells were cultured in RPMI-1640 supplemented with 10% inactivated FBS (iFBS), 2 mM L-glutamine, 100 U/mL penicillin, and 100 μg/mL streptomycin at 37 °C and 5% CO_2_. Seeding of the L6 cells was done in 96-well plates at 4 × 10^3^ cells per well in 100 µL volume; they were then cultured for 24 h. Ten millimolar (10 mM) DMSO stocks of the compounds to be tested were used to prepare 16-point dose-response curves. Each concentration was evaluated in triplicate (using the cell culture medium as diluent) at a final volume of 300 µL (compound concentration range of 50–0.00152 µM and DMSO at 0.5 %). The spent growth medium from the 24 h-seeded cells was carefully replaced with 100 µL of the prepared working concentrations of the compounds to be tested and incubated for 72 h. The spent medium was again carefully replaced (without disturbing the layer of cells at the bottom of each well) with 100 µL fresh medium, after which 20 µL of 0.02% resazurin (previously diluted in 1X PBS) was added to the wells and incubated for 2 h in the dark at 37 °C. The plates were then shaken for 5 s, and fluorescence was read at 570–590 nm in a TECAN Infinite F200 fluorimeter. 

## 3. Results and Discussion

### 3.1. Isolation and Structural Characterization of Compounds

The eight meroterpenoids (**1**–**8**) isolated may be sub-grouped into three sub-chemical classes, namely, the memnobotrin-like pentacyclic lactams (**1**–**3**), the prenylated isobenzofuranones (**4**, **5**), and the cyclic isobenzofuranone analogues (**6**, **7**, **8**). The structural elucidation of each compound is described below, starting with the memnobotrin-like pentacyclic lactams.

Memnobotrin C (**1**) was isolated as a yellowish amorphous solid. Positive mode electrospray ionization time-of-flight mass spectrometric analysis ((+)-ESI-TOF MS) of the compound identified a protonated adduct at *m/z* 428.2440, which corresponded to the molecular formula C_25_H_33_NO_5_ (calcd. for C_25_H_34_NO_5_, 428.2431). The carbon-13 nuclear magnetic resonance (^13^C NMR) spectrum of the compound (Table 1) showed twenty-five carbon signals in total, including two carbonyl carbons at δ_C_ 219.7(C3), 171.7 (C7′) and six *sp*^2^ aromatic carbons at δ_C_ 115.4, 157.9, 100.8, 122.2, 132.6, and 150.0 (C1′–C6′ with oxygenation at C2′ and C6′), which accounted for five degrees of unsaturation in the molecule. These data, in combination with the molecular formula of C_25_H_33_NO_5,_ which indicates a total of ten degrees of unsaturation, showed compound **1** to be pentacyclic. Additionally, the NMR data of **1** showed eight *sp*^3^ methylene groups, one of them oxygenated (C10′ δ_C_ 61.4, δ_H_ 3.78) and another attached to a nitrogen atom (C9′ δ_C_ 46.4, δ_H_ 3.68). There were also two *sp*^3^ methine groups, three *sp*^3^ quaternary carbons with one of them oxygenated (C8 δ_C_ 78.2), and four singlet methyl groups identified. The following intense heteronuclear multiple bond correlations (HMBC) of the four methyl singlets, i.e., δ_H_ 1.27 (H12) with δ_C_ 41.5 (C7), 78.2 (C8), and 52.3 (C9); δ_H_ 1.09 (H13) with δ_C_ 219.7 (C3), 48.6 (C4), 56.1 (C5), and 27.1 (C14); δ_H_ 1.14 (H14) with δ_C_ 219.7 (C3), 48.6 (C4), 56.1 (C5), and 21.9 (C13); and finally δ_H_ 1.09 (H15) with δ_C_ 39.0 (C1), 52.3 (C9), and 37.8 (C10), were consistent with the presence of the drimane-like substructure X in compound **1** (Figure 2). The low-field chemical shifts of the two hydrogens at δ_H_ 2.66, 2.48 placed this *sp^3^* methylene at C2, contiguous to the ketone at δ_C_ 219.7 (C3). The correlation spectroscopy (COSY) between the hydrogen pairs H1/H2 and H6/H7, coupled with HMBC correlations of H2 (δ_H_ 2.66) with C1 and C3, corroborated the drimane-like substructure X of compound **1** (Figure 2). Further, the two downfield hydrogens of the nitrogen-attached methylene at δ_H_ 3.68 (H9′) gave HMBC cross-peaks with the carbons at δ_C_ 171.7 (C7′), 50.2 (C8′), and 61.4 (C10′). The two oxygenated methylene hydrogens at δ_H_ 3.78 (H10′) only gave HMBC cross-peaks with carbon C9′ (δ_C_ 46.4). These correlations, together with those observed from the singlet aromatic hydrogen at δ_H_ 6.73 (H3′) to carbons δ_C_ 115.4 (C1′), 157.9 (C2′), 122.2 (C4′), and 171.7 (C7′), and also the ones from the two methylene hydrogens at δ_H_ 4.36 (H8′) to the carbons at δ_C_ 122.2 (C4′), 132.6 (C5′), 150.0 (C6′) and 171.7 (C7′), led to the identification of the phthalimidine-like substructure Y in compound **1** (Figure 2). The HMBC cross-peaks from the two low-field *sp*^3^ methylene hydrogens at δ_H_ 2.79 and 2.47 (H11) to the aromatic carbons C1′, C2′, and C6′, to the methylene carbon C9 and also to the oxygenated carbon C8 established the pyran ring that connects the phthalimidine-like substructure Y to the drimane-like substructure X and completes the planar structure of compound **1**. This planar structure of compound **1** was confirmed to correspond to a keto-derivative of memnobotrin B previously isolated from *Memnoniella echinata* [17], in which the acetate group attached to C3 in the latter is replaced by a ketone in **1**. Although the overlapping proton (^1^H) NMR signals for the axial hydrogens H5, H7ax and H9 complicated the determination of the relative configuration in **1**, it was assigned using the key NOESY correlations H5/H13 and H9/H11eq, which placed the axial hydrogens H5 and H9, together with the methyl group C13 on the same face of the molecule (Figure 2). On the other hand, H6ax/H15 and H11ax/H15 NOESY correlations oriented the axial methyl C15 on the opposite face of the molecule (Figure 2). Intense NOESY correlations were observed from H12 to H7eq and H11ax, thus establishing a β orientation for methyl C12 and confirming an *R* configuration at C8, opposite to what was previously reported for memnobotrin B [17]. Given the similarity in the NMR data of compound **1** and memnobotrin B and the fact that both compounds have been isolated from the same fungal genus, *Memnoniella*, suggesting analogous biosynthetic routes, the absolute configuration of compound **1** is predicted to be the same as previously determined by X-ray crystallography for memnobotrin A (and by extension, B) [17], excluding the configuration at C8, as already explained above.

(+)-ESI-TOF MS analysis was used to assign a molecular formula of C_25_H_35_NO_5_ to memnobotrin D (**2**) based on the presence of a protonated adduct at *m/z* 430.2595 (calcd. for C_25_H_36_NO_5_, 430.2588). The NMR data of compounds **2** and **1** (Table 1) were very similar, with the main difference being the replacement of the carbonyl ketone at δ_C_ 219.7 in compound **1** with a hydroxylated methine group in compound **2** (δ_H_ 3.21_,_ δ_C_ 79.5), giving rise to the two additional hydrogens in the molecular formula of compound **2** (i.e., reduction of the C3 ketone in compound **1** to a hydroxy methine in compound **2**). The placement of this new hydroxylated *sp*^3^ methine signal at C3 was confirmed by the HMBC cross-peaks of the hydrogen at δ_H_ 3.21 (H3) with carbons C4 (δ_C_ 40.0), C13 (δ_C_ 28.8) and C14 (δ_C_ 16.3). Further, HMBC cross-peaks were observed from hydrogens H13 and H14 to the oxygenated carbon at C3 (δ_C_ 79.5), thus arriving at the planar structure of compound **2** (Figure 2). Since very similar NOESY correlations were observed in both compounds **1** and **2** (Figure 2), the relative and absolute configurations of both compounds are proposed to be identical. It is worth mentioning that in compound **2,** the dispersion of the signals for hydrogens H5, H7ax and H9 allowed the confirmation of their axial orientation via the key NOESY correlations from H5 to H7ax and H9. An axial orientation was also established for the new hydrogen H3 based on the existence of a large axial/axial coupling constant (11.2 Hz) to H2ax and key NOESY correlations between H3 and H2eq, H1ax, H5 and H13 (Figure 2).

Memnobotrin E (**3**) was easily identified due to its striking similarity to the known memnobotrin B [17]. The molecular formula of this compound, C_27_H_37_NO_6_, determined by (+)-ESI-TOF mass spectrometry, was reported for a total of 14 other compounds (including memnobotrin B) in the Dictionary of Natural Products [30]. The proton, carbon-13 and two-dimensional (^1^H, ^13^C and 2D) NMR of compound **3** (Table 2) confirmed its close structural similarity to memnobotrin B except for one difference. As described above for compounds **1** and **2** (memnobotrins C and D), intense NOESY cross-peaks were also observed between the methyl hydrogens H12 and the methylene hydrogen H7eq and also between the methylene hydrogen H11ax and methyl hydrogens H12 and H15, establishing a β orientation of methyl C12 and therefore an *R* configuration at C8 for compound **3**. Memnobotrin E (**3**) was therefore confirmed to be 8-*epi*-memnobotrin B.

With respect to the two prenylated isobenzofuranones isolated (**4**, **5**), compound **4** appeared as a yellowish amorphous solid with a molecular formula of C_29_H_42_O_10_ established on the basis of its (+)-ESI-TOF MS analysis, which showed the presence of an ammonium adduct at m/z 568.3131 (calcd. for C_29_H_46_NO_10_, 568.3116). The NMR data (Table 3) of this compound showed a total of twenty-nine carbons, including one carbonyl carbon at δ_C_ 174.4 (C3), six *sp*^2^ aromatic carbons at δ_C_ 103.9 (C3a), 165.0 (C4 oxygenated), 116.9 (C5), 156.6 (C6 oxygenated), 101.6 (C7), and 148.3 (C7a), and four olefinic carbons at δ_C_ 123.5 (C2′), 135.9 (C3′), 125.5 (C6′), and 136.1 (C7′). These carbons account for six degrees of unsaturation which, when compared to the molecular formula of C_29_H_42_O_10,_ indicates that the compound has three ring systems, one of them being the aromatic ring. A pyran-sugar ring moiety (the second ring) was also readily identified in the molecule by the chemical shifts of the five oxygenated *sp*^3^ methine groups (C1′′–C5′′) and an *sp*^3^ oxymethylene group (C6′′) in the region of 3.18–4.68 ppm, which were, respectively, placed using the multiplicity of the protons in combination with the COSY correlations observed between the following pairs of hydrogens: H1′′/H2′′, H2′′/H3′′, H3′′/H4′′, H4′′/H5′′and H5′′/H6′′. Considering the coupling constants measured for most of the hydrogens in the sugar moiety and the key NOESY correlations from H1′′ to H2′′, H3′′and H5′′, together with the absence of a NOESY correlation between H4′′ and any of the aforementioned hydrogens, the pyran-sugar moiety in compound **4** was identified as β-mannopyranoside. Due to the scarcity of sample, the absolute configuration of the sugar was tentatively proposed as D. An intense HMBC cross-peak between the hydrogen at δ_H_ 4.68 (H1′′) and the carbon at δ_C_ 79.1 (C11′) connected the anomeric carbon (C1′′) of the β-D-mannopyranosyl moiety to the C11′ prenyl part of the molecule via an oxygen bridge. The remaining part of the prenyl chain was established using the intense HMBC cross-peaks of the four singlet methyl hydrogens as follows; δ_H_ 1.21 (H12′) with carbons δ_C_ 42.3 (C10′), 79.1 (C11′), and 27.2 (C13′); δ_H_ 1.22 (H13′) with carbons δ_C_ 42.3 (C10′), 79.1 (C11′), and 26.6 (C12′); δ_H_ 1.55 (H14′) with carbons δ_C_ 125.5 (C6′), 136.1 (C7′), and 41.1 (C8′); and δ_H_ 1.78 (H15′) with carbons δ_C_ 123.5 (C2′), 135.9 (C3′), and 41.0 (C4′). The key COSY correlations shown in Figure 3 supported the structural determination of this part of the molecule. HMBC cross-peaks of the hydrogen at δ_H_ 3.35 (H1′) with carbons C2′, C3′, C4, C5, and C6 linked the prenyl and aromatic parts of the molecule via the C1′-C5 bond. Additionally, HMBC from the singlet aromatic hydrogen at δ_H_ 6.47 (H7) to carbons δ_C_ 103.9 (C3a), C4, C5, and C6 defined the presence of a penta-substituted diol aromatic ring and its connection to the oxygenated methylene carbon at δ_C_ 71.3 (C1). The low-field singlet *sp*^3^ methylene hydrogens at δ_H_ 5.20 (H1) showed HMBC correlations with carbons C3, C3a, and C7 (Figure 3), thus fusing the aromatic ring to the furanone ring. The positions of the oxygenated methylene (C1) and carbonyl (C3) carbons of the furanone ring were confirmed by the intense NOESY and COSY correlations between the hydrogens at H1 and H7. All the above data confirmed compound **4** as a 10′-dehydroxy-11′-β-D-mannopyranosyl C3-carbonyl derivative (rather than C1 carbonyl) of memnoconol [17]. 

In the case of compound **5**, its (+)-ESI-TOF MS analysis showed the presence of a [M+H]^+^ ion at *m/z* 403.2118 (calcd. for C_23_H_31_O_6,_ 403.2115), thus establishing a molecular formula of C_23_H_30_O_6_. There were many similarities between the NMR data of compounds **5** and **4** (Table 3); however, one readily noticeable difference was the absence of the β-D-mannopyranosyl moiety in compound **5**. The ^13^C NMR data of this compound showed a total of twenty-three carbons. These included eleven *sp*^2^ carbons, one carbonyl at δ_C_ 172.4 (C3), six aromatic carbons at δ_C_ 102.2 (C3a), 166.1 (C4 oxygenated), 111.3 (C5), 155.6 (C6 oxygenated), 103.1 (C7), and 151.5 (C7a), and four olefinic carbons at δ_C_ 119.0 (C1′), 127.2 (C2′), 125.7 (C6′), and 136.3 (C7′). These eleven unsaturated *sp*^2^ carbons accounted for six degrees of unsaturation in the molecule, which, when compared to its formula of C_23_H_30_O_6,_ indicates that compound **5** is tricyclic. HMBC cross-peaks observed from the singlet aromatic hydrogen at δ_H_ 6.31 (H7) to the carbons at δ_C_ 70.3 (C1), C3a and C5, together with the HMBC cross-peak of the singlet low-field *sp*^3^ methylene hydrogens at δ_H_ 5.11 (H1) to carbons C3, C3a, C7, and C7a, established a substructure of fused aromatic and furanone rings. NOESY and COSY correlations observed between the hydrogens at H1 and H7 confirmed the placement of carbons C1 and C3 in the same positions as previously observed in compound **4**. Apart from the absence of the β-D-mannopyranosyl moiety, a second difference was identified in the prenyl part of compound **5**, where the singlet methyl group at C15′ gave chemical shifts of δ_H_ 1.45/δ_C_ 27.1 in comparison to the previous observance of this methyl group at δ_H_ 1.78/ δ_C_ 16.4 in compound **4**. Additionally, the hydrogens of this methyl group(H15′) gave HMBC cross-peaks with the carbons at δ_C_ 42.2 (C4′), 81.0 (C3′), and C2′, suggesting the placement of this C15′ methyl group at the oxygenated carbon C3′, resulting in the presence of an ether bridge between C3′ and the aromatic carbon C6 (i.e., C3′-O-C6 cyclization). The C3′-O-C6 cyclization caused the C3′-C2′ double bond previously observed in compound **4** to shift to a new position between C1′ and C2′ in compound **5**, giving rise to a substructure similar to what was previously reported for salfredin B_11_ [31]. This substructure was confirmed by the fact that the NMR chemical shifts reported for salfredin B_11_ were very similar to those recorded for this substructural part of compound **5** [31]. The introduction of a new pyran ring, together with the shifting of the position of the double bond in compound **5,** were further confirmed by two HMBC correlations, i.e., (a) the one from the olefinic hydrogen at δ_H_ 6.72 (H1′) to carbons C3′, C4 and C6, and (b) the other from the olefinic hydrogen at δ_H_ 5.52 (H2′) to carbons C3′ and C5. The last difference between compounds **4** and **5** was at C10′, where a hydroxylation is observed in compound **5** (CH-OH, δ_C_ 79.1 δ_H_ 3.21) as opposed to the CH_2_ group present in compound **4** (δ_C_ 42.3 δ_H_ 1.44). The structure of compound **5** (Figure 3) was therefore determined as a new pyran derivative of a C3-carbonyl analogue of memnoconol [17]. The scarcity of the sample prevented the determination of the absolute configuration at chiral centers C3′ and C10′.

With respect to the third compound class (cyclic isobenzofuranone analogues **6**, **7**, and **8**), the molecular formula of compound **6** was established as C_22_H_32_O_3_ based on the presence of a protonated adduct (i.e., a [M+H]^+^ ion) at *m/z* 345.2426 in its (+)-ESI-TOF MS data (calcd. for C_22_H_33_O_3_, 345.2424). The NMR data of this compound showed a total of twenty-two carbons (Table 4), six of them being *sp*^2^ aromatics at δ_C_ 108.2, 156.5, 108.3, 138.0, 109.8, and 154.8 (C1′–C6′), which together with the ring accounted for four of the degrees of unsaturation. When compared to the molecular formula of C_22_H_32_O_3_, this suggests the compound has three more rings. The intense HMBC cross-peaks observed for the four singlet methyl hydrogens at δ_H_ 1.15 (H12), 1.01 (H13), 0.81 (H14), 0.94 (H15), together with the HMBC cross-peaks from the two low-field *sp*^3^ methylene hydrogens at δ_H_ 2.57, 2.27 (H11) to the aromatic carbons C1′, C2′, C6′, the methine carbon C9 and also to the oxygenated quaternary carbon C8, were used to establish the structure of compound **6** as a drimane-like substructure connected to an aromatic ring through a pyran ring (same as in compound **2**). There was a fifth singlet methyl hydrogen at δ_H_ 2.14 (H7′) in the NMR data, which gave intense HMBC cross-peaks with carbons C3′, C4′ and C5′, hence defining the tetrasubstituted aromatic ring and establishing the planar structure of compound **6** as a tetracyclic benzopyrane (Figure 4). HMBC correlations from hydrogen H3′ to carbons C1′, C2′, C5′ and C7′, and from hydrogen H5′ to carbons C1′, C3′, C6′ and C7′ confirmed this structural proposal. NOESY correlations similar to those observed in compound **2** established the same stereochemistry in both compounds. 

In the case of compound **7**, the presence of a protonated adduct ([M+H]^+^ ion) at *m/z* 385.2014 in its (+)-ESI-TOF MS data (calcd. for C_23_H_29_O_5,_ 385.2010) agreed with a molecular formula of C_23_H_28_O_5_ for the compound. The NMR data of this compound showed a total of twenty-three carbons with many similarities to compound **6** (Table 4), including six *sp*^2^ aromatic carbons at δ_C_ 117.0, 165.6, 101.9, 103.3, 148.2, and 156.6 (C1′–C6′). However, two additional olefinic carbons at δ_C_ 149.5 (C8) and 108.4 (C12) and two carbonyl carbons at δ_C_ 219.0 (C3) and 174.7 (C8′) were identified in the NMR data of compound **7**. In all, these accounted for seven degrees of unsaturation, which, when compared to the molecular formula of C_23_H_28_O_5_, suggests the presence of three cycles in addition to the aromatic ring. Three singlet methyl hydrogen signals were present in this compound, and from these, the following HMBC cross-peaks were observed: δ_H_ 1.04 (H13) to carbons δ_C_ 26.5 (C14), 48.8 (C4), 56.9 (C5), and C3 (ketone); δ_H_ 1.07 (H14) to carbons δ_C_ 22.2 (C13), C4, C5, and C3; and δ_H_ 1.01 (H15) to carbons δ_C_ 38.4 (C1), 41.0 (C10), C5, and 54.4 (C9). These HMBCs defined a drimane-like substructure (with a C3 ketone) for compound **7**. Further, HMBC cross-peaks from the two low-field olefinic hydrogens at δ_H_ 5.11 and 4.73 (H12) to the carbons at δ_C_ 39.2 (C7) and C9, together with HMBC cross-peaks from hydrogens H7, H9 and H11 all to carbon C8, placed a double bond between carbons C8 and C12, thus breaking the C8-O-C6′ bridge and opening up the pyran ring previously observed at this position in compound **6** (Figure 4). Additionally, HMBC cross-peaks similar to those previously described in compounds **4** and **5** established the same substructure for the isobenzofuranone part of compound **7** (C1′ to C8′). The NOESY and COSY correlations observed between hydrogens H3′ and H7′ corroborated the carbonyl position at C8′, thus completing the proposed structure of compound **7** (Figure 4). NOESY correlations around the drimane substructure confirmed the same stereochemistry in both compounds **6** and **7**. 

The structure of compound **8** was easily elucidated due to its close similarity to phomoarcherin A [32]. The molecular formula of compound **8**, determined as C_23_H_30_O_5_ by (+)-ESI-TOF MS analysis, matched that of phomoarcherin A [32]. The ^1^H, ^13^C and 2D NMR data of **8** (Table 5) confirmed it to be very similar to phomoarcherin A with two main exceptions: (a) HMBC cross-peaks similar to those found in compound **7** established the same substructure for the isobenzofuranone part (carbonyl at C8′) of compound **8**, and (b) NOESY correlations (around the drimane substructure) similar to those observed in compound **2** established the same stereochemistry in both compounds **2** and **8**. Thus, as shown in Figure 5, the structure of compound **8** was established as the 8-*epi*, C8′-carbonyl analogue of phomoarcherin A [32].

All the eight new compounds (**1**–**8**) isolated from the *M. dichroa* (CF-080171) extract belong to the tetraketide-terpenoid class of compounds, which are known to be the largest class of meroterpenoids isolated from fungi [9]. A plausible biosynthetic pathway (Figure 1) may involve an initial condensation step between orsellinic acid and farsenyl pyrophosphate to form a linear meroterpenoid intermediate, which may later undergo subsequent derivatizations to produce the various compounds isolated [9]. For example, the glycosylation of the linear intermediate at C11′ and cyclization between the unstable aldehyde and a pre-oxidized methyl substituent of the orsellinic acid residue results in the creation of the isobenzofuranone ring and the eventual production of compound **4** [33,34,35]. Alternatively, hydroxylations at C10′, C11′ and cyclization between C3′ and the para hydroxyl group of the orsellinic acid residue produce compound **5** [36,37]. The cyclic derivatives, **6**–**8**, are formed by total cyclization of the prenyl substructure of the linear intermediate (or partial cyclization in the case of compound **7**) and either lactonization of the orsellinic residue in the case of compounds **7** and **8** or the decarboxylation of orsellinic residue in the case of compound **6**, with different degrees of oxidation in the resulting drimane residue as in each case [38]. The cyclic compounds **1**–**3** may also be formed by total cyclization of the prenyl part of the intermediate followed by lactamization (instead of lactonization) of the orsellinic acid residue with further derivatizations [39]. The resulting drimane residue could also undergo further derivatization, as in each case [9]. 

### 3.2. Biological Activity

For early drug discovery purposes, the standard *P. falciparum* 3D7 lactase dehydrogenase and transgenic *T. cruzi β-*D-galactosidase whole antiparasitic assays were utilized in the bioassay-guided process to purify the active components of the *M. dichroa* CF-080171 extract, which had been previously identified as active against both parasites. All the isolated compounds (**1**–**8**) were tested in their pure forms against the *P. falciparum* 3D7 and *T. cruzi* Tulahuen C4 parasitic strains, and their EC_50_ values were determined (Table 6). Of the different compound classes isolated, memnobotrins C (**1**) and D (**2**) showed the most potent antiparasitic activity with EC_50_ values of 0.040 and 0.201 µM, respectively, against *P. falciparum* 3D7, and 0.226 and 1.37 µM respectively, against *T. cruzi* Tulahuen C4. The presence of a ketone at C3 in compound **1** improves its biological activity 5-fold against both *P. falciparum 3D7* and *T. cruzi* Tulahuen C4 when compared to the presence of a hydroxyl at this same position (C3) in compound **2**. However, memnobotrin E (**3**), which has an acetate group attachment at this same position (C3), showed 179-fold lower potency against *P. falciparum* 3D7 and 157-fold lower potency against *T. cruzi* Tulahuen C4 in comparison to compound **1** (Table 6). 

In the case of compounds **4** and **5**, which had some structural similarities to memnoconol [17], compound **4** exhibited an interesting potency with EC_50_ values of 0.243 and 0.934 µM against *P. falciparum* 3D7 and *T. cruzi* Tulahuen C4 respectively, whereas compound **5** showed moderate potency against *P. falciparum* 3D7 (EC_50_ of 8 µM) and was inactive against *T. cruzi* Tulahuen C4 at 50 µM. The presence of a sugar moiety in compound **4** may be implicated in the significantly higher bioactivity observed in this compound. With respect to compounds **6**–**8**, only **6** showed moderate activity in both parasites with EC_50_ values of 4.0 and 7.0 µM against *P. falciparum* 3D7 and *T. cruzi* Tulahuen C4, respectively. Compound **7** was inactive in both parasites at 100 and 50 µM, respectively, whereas compound **8** showed slight activity (EC_50_ of 17.8 µM) against *P. falciparum* 3D7 only.

In order to determine their general cytotoxic effect, the EC_50_ values of the isolated compounds were also determined in vitro against two different cell lines, i.e., liver carcinoma HepG2 cells and L6 rat skeletal muscle cells. Although the crude *M. dichroa* CF-080171 extract from which the compounds were isolated had previously been cleared as non-cytotoxic against the HepG2 cells at the primary screening stage, it was essential to confirm the potential cytotoxic effect of the isolated pure compounds since factors such as purity and failure to reach the effective inhibitory concentration may have masked the proper effects of the respective compounds in the crude extract prior to their purification. The L6 rat skeletal muscle cells were used here as a second line of cytotoxic screen to ensure robustness since the intracellular *T. cruzi β-*D-galactosidase antiparasitic assay is performed in this same host cell line. As seen in Table 6, the three new compounds which exhibited the most interesting antiparasitic activity, i.e., compounds **1**, **2** and **4**, also exhibited low micromolar EC_50_ values of 1.20, 4.53, and 4.84 µM, respectively, when tested for their cytotoxicity against hepatocytic carcinoma Hep G2 cells by means of a cell viability MTT assay. Although the three compounds (**1**, **2** and **4**) demonstrated some level of selectivity towards the *P. falciparum* 3D7 (selectivity indices of 30, 22.5 and 19.9, respectively) as compared to their cytotoxicity against the Hep G2 cells, the fact that the EC_50_ values recorded were in the low micromolar range was a hint of general cytotoxicity. To confirm this, the two most potent compounds, **1** and **2**, were further tested against L6 rat skeletal muscle cells, and both proved to be cytotoxic with EC_50_ values of ˂0.098 µM for compound **1** and 1.39 µM for compound **2**. The L6 host cells are routinely used in evaluating cytotoxicity of potential antimalarial compounds [40,41,42]; thus, the cytotoxic nature of compounds **1** and **2** in this cell line demonstrates their inherent cytotoxic nature. As would be expected, compound **6,** which showed moderate antiparasitic activity in both parasites, also exhibited some cytotoxicity against L6 cells (EC_50_ value of 2.0 µM), whereas the *T. cruzi*-inactive compounds **5** and **7** were also inactive at 50 µM in L6 host cells (Table 6). 

Although the literature reports many triprenyl phenolic compounds isolated from *Stachybotrys* and *Memnoniella* fungi, only two compounds belonging to the memnobotrin subclass have ever been previously isolated, i.e., memnobotrins A and B. Memnobotrin B was reported to show some cytotoxic activity at 100 µM against three different cell lines, with inhibition percentages in the range of 80–90% [17]. Considering these observations, the newly discovered antiparasitic memnobotrins C and D (compounds **1** and **2**) provide very interesting biologically relevant analogues to the memnobotrin subclass despite their inherent cytotoxicity. Additionally, the superior biological activity of memnobotrins C and D in comparison to memnobotrins A, B and E, shows how the C3-substituent affects the overall potency of this compound class, thus providing some insight into their structure-activity relationship (SAR) with respect to possible chemotherapeutic developments using medicinal chemistry methods. Some of these medicinal chemistry methods may include (a) the full or partial direct replacement of the C3 substituent to generate several classes of analogues, (b) the reduction of the electronic densities around certain strategic parts of the molecule, and (c) the introduction of a structural element of metabolic interest [43]. The various analogues generated from such medicinal chemistry methods could be tested in a SAR study to ascertain which of them would exhibit the best selectivity against only the parasites for further development. It would also be interesting to further investigate the biological effect of this compound class in (a) cell life cycle/morphological perturbations and (b) target deconvolution studies in both *P. falciparum* and *T. cruzi* parasites [44].

## 4. Conclusions

Bioassay guided-purification of an extract produced by the fungus *M. dichroa* CF-080171 led to the isolation of eight new meroterpenoid compounds (**1**–**8**) of varying antiparasitic potencies against *P. falciparum* 3D7 and *T. cruzi* Tulahuen C4 whole parasites. The absolute structures of all the compounds were established by a combination of (+)-ESI-TOF high-resolution mass spectrometry and nuclear magnetic resonance spectroscopy. Compounds **1** and **2**, which belong to the memnobotrin subfamily of triprenyl meroterpenoids, were the most potent compounds, demonstrating antiparasitic activity in the nanomolar range against both parasites (EC_50_ 0.04–1.37 µM). Compound **4**, a glycosylated tricyclic isobenzofuranone, also exhibited interesting biological activity against both parasites (EC_50_ values 0.243 and 0.934, respectively), whereas the tri/tetra/pentacyclic isobenzofuranones (**5**, **7**, **8**) and the tetracyclic benzopyrane (**6**) showed low or no biological activity against both parasites. These newly isolated meroterpenoids from *M. dichroa* CF-080171 further demonstrate the huge potential of this fungal natural source as a dependable biofactory for the isolation and characterization of novel biomolecules with possible chemotherapeutic applications. 

## Data Availability

Sequences of the complete ITS1-5.8S-ITS2 and initial 28S region or independent ITS and partial 28S rDNA of *M. dichroa* CF-080171 were deposited in GenBank, with the accession number OP554575.

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
