# Peer review of "Antiparasitic Meroterpenoids Isolated from Memnoniella dichroa CF-080171"

_pharmaceutics, 2023, doi:10.3390/pharmaceutics15020492_

Round 1
Reviewer 1 Report (New Reviewer)
This manuscript using bioassay guided-purification of an extract produced by the fungus M. dichroa CF-080171 isolated eight new meroterpenoid compounds. The structures of the compounds were identified by a combination of ESI-TOF and NMR. This revised manuscript organized very well and described very clearly. In my opinion, this manuscript could be accepted after minor modification.
In table 6, EC50 for compounds 1-8, should the data be displayed as average ± SD, rather than +SD?
Author Response
We acknowledge the effort of reviewer #1 in reviewing our manuscript and thank him/her for the feedback. We have duly updated and uploaded a revised version of the manuscript in which this change has been made (table 6).
Reviewer 2 Report (New Reviewer)
In this manuscript, the isolation and structure elucidation of Meroterpenoids from Memnoniella dichroa have been reported. Also, the antiprotozoal activity of isolated compounds was tested.
The researchers claimed (section 2.4) that the bio-guided assay fractionation technique has been used for tracking bioactive compounds, but I did not find any methodology in the text. please clarify this issue.
I am not convinced with your conclusion about the absolute configuration of compound 1 and direct comparison with memnobotrin B. In conclusion, you need to compare specific optical rotations using circular dichroism data. Also, for other compounds use same protocol.
Author Response
We acknowledge the effort of reviewer #2 in reviewing our manuscript and thank him/her for the feedback. Regarding the “bio-guided assay fractionation technique”, we have explained in section 2.4 that “the organic phases were combined and dried to yield 892 mg of a crude that tested positive against both, P. falciparum and T. cruzi parasites”. We also further explained in that same section that “Repeated injections ….. in combination with antiparasitic testing …. identified the fractions of interest”. Hence, the DCM extract and all fractions obtained in the different chromatographic (HPLC) runs were tested in the antiparasitic assays where the activity of the original extract was detected. Please note that in terms of fractionation, there is really no difference between a bioassay-guided process and a normal process. It is the testing of the different fractions, followed by purification of the active fractions, that makes the process bioassay-guided.
Regarding the comments on the absolute configuration we agree that obtaining the circular dichroism data of compound 1 in combination with the comparison between its specific optical rotations and that of memnobotrin B would have conclusively determined the absolute configuration of the compound (and that of all the other molecules described). However, in this case we obtained very low amounts of the compounds and this did not permit us to obtain their circular dichroism data, so, we went ahead to compare the isolated compounds to other members of the same structural class previously described in the literature and made absolute configuration predictions based on NOESY experiments in combination with biosynthetic considerations. Note that the absolute configurations indicated in the manuscript are only predictions supported by the NOESY experimental data. Moreover, the ECD technique is not available in our lab and determination of the absolute configuration by this process would have required sending the compounds to external collaborators, a process in which we would have lost track of the necessary care required to handle such low amounts of material.
Reviewer 3 Report (New Reviewer)
This article is interesting, fits the journal's scope, and meets the minimum requirements to be published. However, with the intention that the work can achieve good quality to communicate the information efficiently to the readers, I recommend refining the tone of the writing, particularly in the results and discussion section, so that it is not overly colloquial.
Author Response
We acknowledge reviewer #3´s comments and have refined the tone of writing in the revised manuscript at the sections indicated.
Reviewer 4 Report (New Reviewer)
Dear authors, first I would like to congratulate you for your research. However, the introduction is very superficial, as there is only one paragraph talking about parasites. In the first paragraph, the authors should explain about the parasites, as was done, in a second and third paragraph, they should deepen about the two parasites, bringing their main characteristics, etiological agents, epidemiology, treatment, as well as recent studies in the area.
I missed the connection between the paragraphs. They are loose. No nexus. Use connectives. Use the main articles in the area.
The methodology is very clear and objective, as requested by the Journal.
The title of the work focuses on the antiparasitic activity Meroterpenoids Isolated from Memnoniella dichroa, however, the article has a more focused focus on phytochemistry, with biological activities being complementary. This can be seen in the results, where there was disregard for the part of biological activities, having only 4 paragraphs. When compared to phytochemistry. For the manuscript to be accepted, it is necessary to make such changes.
Author Response
We acknowledge the effort of reviewer #4 in reviewing our manuscript and thank him/her for the feedback. We have included some more information in the introduction, results and discussion parts of the manuscript and have also paid particular attention to the connection between the different paragraphs.
We particularly added some more information in the 3.2. Biological activity part of the Results and discussion section. Since the manuscript describes the detailed structural elucidation of 8 compounds, all of which have been subjected to the same biological testing in four assays, it could be appreciated that the results and discussion of the chemistry part of the work done would be substantially more than the biological aspect of this work.
Reviewer 5 Report (New Reviewer)
This manuscript reports the bioassay guided-isolation of metabolites from Memnoniella dichroa CF-080171 extract, which provided eight new meroterpenoids. The structures of eight compounds were studied by the LC-MS (mass spectrometry) and nuclear magnetic resonance spectroscopy (NMR). The potencies of these compounds were characterized with P. falciparum 3D7, T. cruzi Tulahuen C4 and HepG2 tumoral liver cells. The methodology in this manuscript is interesting. Detailed MS, NMR and potency characterization data are provided. This manuscript can be considered for publication after minor revisions.
Minor points:
1. Page 2 Line 54, the bioassay guided-isolation is a key component in the manuscript for discovering eight compounds. I suggest adding related data to better illustrate this process, such as chromatogram.
2. Authors used MS and NMR as major tools to elucidate structures of isolated compounds. However, only intact mass was utilized in identifying the molecular formula. Did you use other MS based information, such as isotopic pattern and MS/MS fragmentation?
Author Response
We acknowledge the effort of reviewer #5 in reviewing our manuscript and thank him/her for the feedback.
(1) We have now included a schematic of the bioassay-guided isolation process in the supplementary information (Figure S1).
(2) As could be seen in section 2.4. Bioassay-guided isolation of compounds 1-8 (page 5 lines 165 – 206), and the supplementary data (Figures S3, S13, S22, S33, S42, etc.), we also used the dimeric and trimeric ionized species of each of the compounds in confirming their molecular formula determinations.
This manuscript is a resubmission of an earlier submission. The following is a list of the peer review reports and author responses from that submission.
Round 1
Reviewer 1 Report
The manuscript by Annang and collaborators shows the purification of compounds from the Memnoniella dichroa fungus and the bioactivity of such molecules on P. falciparum, T. cruzi, muscle, and tumor cells.
I analyzed the biological part, since it is my area of expertise. In general, biological experiments were poorly described, and this part needs special attention. For example, if I try to repeat such experiments, I will be unable to, because it was not well described. I suggest that you insert the following items: 1) number of cells used in infection; 2) number of P. falciparum used; 3) what parasitic form was used to infect erythrocytes? 4) medium used? 4) Blanks used? 5) Standard drug used? 6) How was EC50 calculated? These same questions work for T. cruzi! Why don't you study intracellular forms of T. cruzi? Such tests are essential for analyzing the efficacy of your molecules. Transgenic parasites were used, but how were they generated? What about the activity of purified molecules on wild parasites?
The authors studied cytotoxicity using two cell lineages: HepG2 and L6 muscle cells. Why? Although the method is slightly different, in essence they are equal. So, there is no need to put them in different subsets.
How many repetitions were performed in biological experiments? It is mandatory to add standard deviation to the data displayed in Table 6. Why were only SIs estimated for P. falciparum? Why not for T. cruzi? Why did the authors use only data related to HepG2 cells to estimate SI? In particular, why used HEPG2 cells for SI, considering that it is a tumor cell lineage that did not display normal physiology?
Although the authors used colorimetric tests to analyze molecule efficacies, the manuscript will improve a lot if pictures showing low parasitism after treatment were added.
Minor: I believe that the introduction lacks the main facts about malaria and American trypanosomiasis; data that can be found in WHO, CDC, etc.
Reviewer 2 Report
In this manuscript, the authors are reporting the isolation, purification, structures elucidation of eight new compounds from the fungus Memnoniella dichroa CF-080171 crude extract. The chemical structures of the new compounds were established on the basis of HRMS and NMR spectroscopy, and the relative stereochemistry of the isolated compounds were established by NOESY experiments. In addition, the authors have also evaluated the isolated compounds for their antiparasitic (P. falciparum 3D7 and T. cruzi Tulahuen), and cytotoxic (HepG2 tumoral liver) activities. Overall, the data are well documented/and discussed, however some major revisions are required in this manuscript prior for its publication as summarized below:
- The authors have mentioned/discussed in several places that they have used the bioassay guided isolation? And throughout the manuscript it was not clear how the authors used this bioassay isolation method to obtain the bioactive compounds (several of them were not active)?
- Page 2, lines 46-50: “Bioassay-guided fractionation” - this section need to be revised, it is not bioassay-guided ….it is just HPLC purification.
- Pages 2-3: General Experimental Procedures: The authors should include the source of parasites/cell lines used in this study (P. falciparum 3D7 and T. cruzi Tulahuen and HepG2 and L6)?
- Lines 99-115: Bioassay-guided of compounds 1-8: This section needs to be rephrased and described in detail. Lines 100-102: How the authors extracted 1L of aqueous with acetone (both are miscible?) Also, the extract was extracted with DCM (only one solvent), followed by HPLC purification, no bioassay guided isolation here, just one extract shows the activities. Also, what is the total amount of the obtained extract?
- The isolated compounds were in the range of 1.0 mg – 0.2 mg? how the authors obtained all these experimental data (optical rotation, NMR, IR, bioactivities with a compound of 0.2 mg? please explain.
- Line 119: “428.244” revise as “428.2440” use four-digit numbers for HRMS.
- Figure 2: COSY between CH2-9’/CH2-10’ is missing in the structure (fragment Y)?
- I recommend the authors to combine all the 2D structures in one single figure of all the compounds (i.e. combine all figures 2-5)
- Pages 6-15 (Lines 194-411), Isolation and structural characterization of compounds: The authors have highlighted in several places the term “absolute configuration”, all the compounds stereochemistry are just relatives not absolute? The authors just used the NORSY experiments, and no attempts to confirm the absolute stereochemistry (e.g. Mosher method, ECD/ECD calculations, comparison with synthetic standard, etc.).
- Tables 1-5: “NMR spectroscopic data (500 MHz, CD3OD) of…” revises as “1H (500 MHz) and 13C (125 MHz) NMR spectroscopic data of…”
- The authors should carefully check the NMR data/multiplicities: e.g. Table 2, “1.63, dd”, the coupling constants are missing?
- References should be checked/formatted according to the journal formats/style, for example, reference 1 “https://doi.org/10.1007/s11101-014-9365-1” but reference 2 “doi: 10.1080/1369378031000137350”, the “doi” are in different formats.
- Supplementary Data:
- The IR spectra are missing and should be included.
- The NMR chemical shifts are missing on all the 1H and 13C NMR spectra and should be included. Also – the authors should show the full spectrum range 1H NMR spectrum (0-12 ppm), 13CNMR (0-220 ppm).
- Figures of antiparasitic and cytotoxicity data for the tested compounds should be included in supplementary material.
Reviewer 3 Report
Manuscript ID: pharmaceutics-1968861 entitled “Antiparasitic Meroterpenoids Isolated from Memnoniella di-2 chroa CF-080171” represents a well-done research work reporting eight new meroterpenoids from the fungal strain Memnoniella di-2 chroa CF-080171, having antiparasitic activity. Structures of the new compounds were assigned on the bases of (+)-ESI-TOF high resolution mass spectrometry and nuclear magnetic resonance spectroscopy. Biologically, compounds 1, 2 and 4 exhibited potent antiparasitic activity against P. falcipa-28 rum 3D7 (EC50 0.04–0.243 μM) and T. cruzi Tulahuen (EC50 0.266-0.934 μM) parasites, as well as cy-29 totoxic activity against HepG2 tumoral liver cells (EC50 1.20-4.84 μM).
I think the work is interest and can be published in Pharmaceutics. However, there are several remarks should be taken in consideration before taking a decision:
1) Although of the interest of the obtained new meroterpenoids, they were afforded in very low quantities (ranged between 0.2 mg to 1.0 mg as maximum), with low quality of purity as shown in the NMR spectroscopy charts (as shown in the supplementary file).
2) This low and insufficient quantities prevented the authors from the determination of the absolute configurations of the eight compounds.
3) In the experimental section, the weight of the fungal extract obtained was not mentioned.
4) The samples of the reported compounds were not quietly dried before applying to NMR measuring, so that the methanol signal and corresponding water signals shown in the 1H NMR spectra are very high, decreasing the compounds signals intensities and their resolutions as well.
5) Compounds 2 and 3 look like a mixture of two stereoisomers as shown obviously in the 13C NMR spectrum! As all compounds reported have strong absorbing chromophores in UV, the authors are required to determine the absolute configuration of the reported compounds using ECD spectroscopy and compare that with the calculated results to find out their absolute configurations.
6) Compound 5 is highly impure as shown in the 1H and 13C NMR spectra!
7) The water signal in the 1H NMR of compound 6 is very up normal and hence its 13C spectrum is not good displaying the compound to be as a mixture of two stereoisomers!
8) The same problem shown for compounds 7 and 8.
9) Off course the purity of the compounds has a great influence on their biological activity, and it will be not correctly decided to which of the compounds the reported antiparasitic activity or cytotoxicity are attributed!

Round 2
Reviewer 1 Report
Dear authors,
I still noted that methodologies need improvement, for instance, how about blank wells? You did not describe it; please go back to the first round of revision and make adequate changes.
As you mentioned, this is a preliminary manuscript, and I think this is not ready to be published in Pharmaceutics. In spite of being preliminary, it is essential to demonstrate morphologically that your molecules decreased intracellular parasitism. In the same scenario, both forms of T. cruzi should be assayed, because any early or late drug discovery program can demonstrate that a given prototype drug is active (even in vitro) on acute and chronic forms of American tripanossomiasis.
Wild parasites should also be tested.
As you detailed the methods, I observed that you use different methods to analyze cytotoxicity, that have different sensitivities: MTT versus resazurin. In this specific case, more details need to be added: type of plates, volumes, blanks, etc. See my first report.
Once again, your introduction seems to be a background on the theme of a short communication manuscript. You need to detail much more about both neglected tropical diseases.
Reviewer 2 Report
The authors have considered/responded to the most of my recommended edits/comments in the revised version of the manuscript, and I recommend the manuscript to be published in the present form.
Reviewer 3 Report
Manuscript ID: pharmaceutics-1968861 entitled “Antiparasitic Meroterpenoids Isolated from Memnoniella dichroa CF-080171” reported the isolation and structure identification of eight new meroterpenoids from the fungal strain Memnoniella dichroa CF-080171, having antiparasitic activity. Structures of the new compounds were assigned on the bases of (+)-ESI-TOF high resolution mass spectrometry and nuclear magnetic resonance spectroscopy. Biologically, compounds 1, 2 and 4 exhibited potent antiparasitic activity against P. falciparum 3D7 (EC50 0.04–0.243 μM) and T. cruzi Tulahuen (EC50 0.266-0.934 μM) parasites, as well as cytotoxic activity against HepG2 tumoral liver cells (EC50 1.20-4.84 μM).
The research idea of the reported manuscript is well however, the quality, purity and insufficient material of the reported compounds are non-acceptable which off course affect on the reported antiparasitic activity, and structural assignments. Moreover, I have given the authors a great chance when I recommended to accept it after major revision to give more time to do the shown below referred points. However, they did not give any response on these points or did such recommended inquiries professionally:
1) Although of the interest of the obtained new meroterpenoids, they were afforded in very low quantities (ranged between 0.2 mg to 1.0 mg as maximum), with low quality of purity as shown in the NMR spectroscopy charts (as shown in the supplementary file).
2) This low and insufficient quantities prevented the authors from the determination of the absolute configurations of the eight compounds.
3) The samples of the reported compounds were not quietly dried before applying to NMR measuring, so that the methanol signal and corresponding water signals shown in the 1H NMR spectra are very high, decreasing the compounds signals intensities and their resolutions as well.
4) Compounds 2 and 3 look like a mixture of two stereoisomers as shown obviously in the 13C NMR spectrum! As all compounds reported have strong absorbing chromophores in UV, the authors are required to determine the absolute configuration of the reported compounds using ECD spectroscopy and compare that with the calculated results to find out their absolute configurations.
5) Compound 5 is highly impure as shown in the 1H and 13C NMR spectra!
6) The water signal in the 1H NMR of compound 6 is very up normal and hence its 13C spectrum is not good displaying the compound to be as a mixture of two stereoisomers!
7) The same problem shown for compounds 7 and 8.
8) Off course the purity of the compounds has a great influence on their biological activity, and it will be not correctly decided to which of the compounds the reported antiparasitic activity or cytotoxicity are attributed!
So, I do not recommend accepting the reported manuscript for publication in Pharmaceutics